# Polydopamine Modified Graphene Oxide-TiO₂ Nanofiller for Reinforcing Physical Properties and Anticorrosion Performance of Waterborne Epoxy Coatings

**Shuli Wang [1], Jiankang Zhu [1], Yongchao Rao [1,*], Beibei Li [2], Shuhua Zhao [1], Haoran Bai [1] and Jiawei Cui [1]**

[1]   School of Petroleum Engineering, Changzhou University, Changzhou 213100, China
[2]   School of Chemistry and Chemical Engineering, Yulin University, Yulin 719000, China
*   Correspondence: ryc@cczu.edu.cn; Tel.: +86-159-6129-6997

**Abstract:** Nano-polydopamine-graphene oxide-TiO₂ (nano-PDA@GO-TiO₂) composites were prepared by dopamine modified graphene oxide (GO) and loaded nano-TiO₂ on the surface of GO. The structure and morphology of nano-PDA@GO-TiO₂ composites were characterized by X-ray diffraction (XRD), Fourier transform infrared spectroscopy (FT-IR), Raman, X-ray photoelectron spectroscopy (XPS), Scanning electron microscope (SEM), and Transmission electron microscope (TEM) Results demonstrate that the introduction of dopamine to functionalize the GO could self-polymerize polydopamine (PDA) on the surfaces of the GO and endow abundant chemical groups reduce the GO. The interaction between the GO and nano-TiO₂ particles could prevent graphene nanosheets from restacking and nano-TiO₂ particles from agglomeration. Nano-PDA@GO-TiO₂ composite material was used as the nano-filler, and nano-PDA@GO-TiO₂ composites waterborne epoxy resin coatings (PGT/WEP) were prepared by dispersing a different content of nano-PDA@GO-TiO₂ composites into waterborne epoxy resin with the help of ultrasonic dispersion and mechanical agitation. The physical properties of PGT/WEP coatings, such as hardness, impact resistance, and adhesion, were tested and the electrochemical performance was evaluated. The results show that dispersing 2% nano-PDA@GO-TiO₂ composites in waterborne epoxy resin could significantly improve the physical properties and corrosion resistance of waterborne epoxy resin coating when compared with pure waterborne epoxy coating.

**Keywords:** dopamine; graphene oxide; nano-TiO₂; composite materials; nanofiller; epoxy coating; physical properties; corrosion resistance

## 1. Introduction

It is known that the damage of petrochemical equipment, oil and gas pipelines, and other infrastructure caused by corrosion of metals, has caused severe waste of resources, environmental pollution, and potential safety problems. One of the most cost-effective ways to prevent metal corrosion is the physical barrier coating, which creates an effective barrier between the metal substrate and the corrosive environment [1]. Epoxy resin coatings have been widely been in the coating industry due to these stable chemical properties, excellent electrical insulation rot, low cure shrinkage, high tensile strength and strong adhesion, and as the most effective method to inhibiting corrosion of the metal substrate [2,3]. However, the traditional epoxy resin anticorrosive coatings are typical solvent-based coating systems, which contain a certain proportion of volatile organic compounds (VOCs) that cause health hazards and environmental problems. In recent years, many countries

have legislation restricting emissions of VOCs [4,5]. Therefore, it is imperative to develop low-cost, low-viscosity, easy-to-clean, high-performance, and environmentally-friendly water-based coatings to replace solvent-based epoxy coatings.

Waterborne epoxy coatings have been extensively used due to their excellent adhesion, chemical resistance, and low shrinkage. And have been commercialized for nearly half a century [3,6]. However, the traditional waterborne epoxy resin system still has some defects. Waterborne epoxy resins usually contain a large amount of hydrophilic groups and surfactants, so that a large number of microporous channels are formed after curing. It shows more permeability to corrosive substances (ie $H_2O$, $O_2$, and $Cl^-$) and it cannot provide long-term corrosion protection for metal substrates [7]. Therefore, the application of waterborne epoxy resin in the coatings market is restricted. In order to address these problems, many researchers have begun to apply nanotechnology to waterborne epoxy coatings to improve the corrosion resistance of waterborne coatings. Graphene and graphene derivatives can significantly improve the resistance of waterborne epoxy coatings due to their large specific surface area, high mechanical properties, good thermal stability, and outstanding barrier effect. Corrosion performance and other properties have become a research hotspot in the coating industry [8,9]. Despite the great breakthroughs in the field of graphene-based waterborne epoxy coatings, there are still some shortcomings in the application of graphene. For example, the strong van der Waals force and high specific surface area cause it to easily aggregate together, ca n't to form a stable chemical bond with water and organic matter and the interface bonding force with the resin decreases, which limit their applications in coatings [10]. It is found that the inorganic nanomaterials, $SiO_2$, $TiO_2$, $Fe_3O_4$, ZnO, etc., are supported on the surface of graphene oxide to prepare a graphene-inorganic nanocomposite with synergistic effect. Depositing nanoparticles not only can prevent graphene oxide nanosheets from restacking, but also endow a new functionality to graphene oxide [11–13]. At present, graphene-inorganic nanocomposites have been widely used in photocatalysts, electronic sensors, and biomedical fields [14–16], and the use of graphene-inorganic nanocomposites as fillers can also have a synergistic effect on improving the corrosion resistance of the coating. Zuo et al. [17] reported that nano-ZnO loaded graphene oxide/epoxy composites were successfully prepared by loading nano-ZnO on the surface of graphene oxide and compounding with epoxy resin. Through a series of characterizations, it is found that nano-ZnO can be uniformly dispersed on graphene oxide, which not only does not change the structure of graphene oxide, but also improves the agglomeration of graphene oxide and also reduces the hydrophilicity of graphene oxide. Compared with epoxy materials, the thermal stability and mechanical properties of nano-ZnO loaded graphene oxide/epoxy composites have been significantly improved. Yu et al. [11] found that used KH550 modified nano-$TiO_2$ on the surface of GO to synthesized $TiO_2$-GO composite material, which could effectively improve the corrosion resistance of the epoxy resin by dispersing in epoxy resin coating. In order to further improve the compatibility of GO and resin, enhance the macroscopic properties of GO/epoxy composites. Using an organic reducing agent (polyethylene glycol [18], isocyanate [19], dopamine [20], etc.) to interact with the reactive functional groups on the surface GO, through the intercalation and coating of the GO sheet to restraining agglomeration, and functionalization of GO [21], which has been confirmed as an effective method to enhance the interaction between GO and resin matrix and improve the compatibility between GO and resin. At the same time, making interfacial adhesion between the GO and polymer composites is improved and the macroscopic properties of the GO/polymer composite are enhanced. Among them, dopamine, as a green environmentally friendly reducing agent, can be easily deposited on the surface of many organic and inorganic substrates by self-polymerization and covalent bonding to form a polydopamine layer. Moreover, abundant reactive functional groups (such as catechol, amines, and imines) in dopamine and polydopamine can be used as active sites for covalent modification with desired molecules [22–24]. Hu et al. [21] found that DA could effectively eliminate the labile oxygen functionality of GO and generate polydopamine functionalized graphene oxide (PDA-GO), because of DA would be oxidized and undergo the rearrangement and intermolecular cross-linking reaction to produce polydopamine (PDA), which would improve the interfacial adhesion

between GO and epoxy, and further beneficial for the homogenous dispersion of GO in epoxy matrix to enhance the thermal conductivity, storage modulus, and mechanical properties of epoxy. Cui et al. [25] reported an eco-friendly anticorrosive water-borne epoxy coating incorporated with the dopamine functionalized GO nanosheets, and found that the GO-PDA nanosheets had superior compatibility with waterborne epoxy matrix. The electrochemical results showed that the corrosion protect ion performance of GO-PDA/EP was improved with respect to the blank EP and GO/EP, owing to the strong interfacial bonding between GO-PDA and EP matrix.

In this paper, dopamine was used to covalently modify graphene oxide to form a polydopamine (PDA) layer with excellent adhesion with GO surface, and nano-$TiO_2$ particles were firmly deposited on the surfaces of the graphene oxide (GO) via the electrostatic and hydrogen interactions forming nano-PDA@GO-$TiO_2$ composites. Adding it to the waterborne epoxy coating with the help of ultrasonic dispersion method, and systematically studied the basic physical properties and corrosion resistance of the nano-PDA@GO-$TiO_2$ composites with different contents on the impact resistance, hardness, and adhesion of the waterborne epoxy coating. The effects of corrosion resistance and corrosion resistance of nano-PDA@GO-$TiO_2$ composites waterborne epoxy coatings (PGT/WEP) were also investigated by electrochemical experiments.

## 2. Materials and Methods

### 2.1. Materials

Graphene oxide was purchased from Jiangnan Graphene Research Institute; Nano-$TiO_2$ (99.8%, 5–10 nm, anatase), tris(hydroxymethyl)aminomethane (99.9%) and dopamine hydrochloride (98%) were obtained from Shanghai Aladdin Biotechnology Co., Ltd. Company (Shanghai, China); Waterborne epoxy resin H228A and curing agent H228B are industrial grade and were provided by Shanghai Han Zhong Coating Co., Ltd(Shanghai, China).; Sodium chloride (AR), acetone (AR), and anhydrous ethanol (AR) were supplied by Sinopharm Chemical Reagent Co., Ltd.( Beijing ,China)

### 2.2. Preparation of PDA@GO-$TiO_2$ Nanocomposites

The preparation method of nano-PDA@GO-$TiO_2$ composites material is referred to [26–28], and optimized on the basis of them. The specific preparation process is shown in Figure 1 s. First, 0.10 g of graphene oxide was dispersed in 450 mL Tris buffer (10 mM, pH = 8.5) and continued ultrasonic for 45 min. to obtain graphene oxide dispersion, then 0.50 g of dopamine hydrochloride and 0.30 g of nano-$TiO_2$ particles were added into the dispersion. The mixed solution was heated at 45 °C for 24 h without any disturbance, and the solution gradually changed from yellowish brown to black. After the reaction was completed, the mixed system was centrifuged, and the centrifuged product was repeatedly washed with ethanol and deionized water several times to remove unreacted dopamine and other reagents. And freeze dried under vacuum to a constant mass to obtain the final product nano-PDA @GO-$TiO_2$ composite.

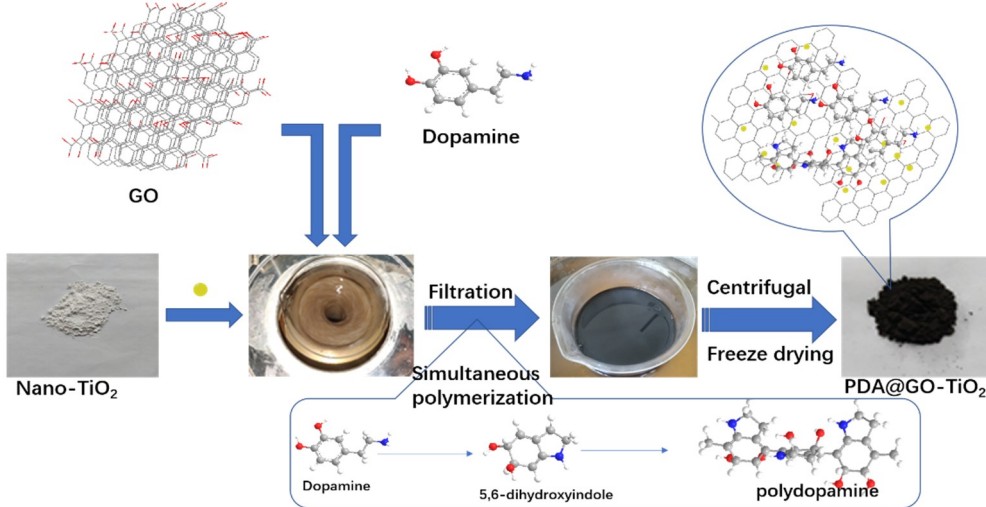

**Figure 1.** Schematic diagram of the preparation process of nano-polydopamine (PDA)@graphene oxide-TiO$_2$ (GO-TiO$_2$) composites.

### 2.3. Preparation of Nano PDA@GO-TiO$_2$/Waterborne Epoxy Coatings

In order to remove the surface rust, of the Q235 steel. Firstly, sanding the surface of the Q235 steel test piece with coarse sandpaper and fine sandpaper, respectively. Then cleaning the polished test piece with deionized water. Finally, wiping the surface of the test piece with absolute ethanol and acetone, respectively.

The specific preparation process of the composite coating is shown in Figure 2. Nano-PDA@GO-TiO$_2$ composite (0.20 g) was dispersed in 20 mL of deionized water, by ultrasonication for 30 min at room temperature to obtain nano-PDA@GO-TiO$_2$ aqueous dispersion. Then, amount of waterborne epoxy resin was added into nano-PDA@GO-TiO$_2$ aqueous dispersion, and the mixture was dispersed homogeneously by ultrasonication for 30 min at the room temperature. Then the precalculated water borne curing agent was added to the mixture and stirred with magnetic stirrers for 30 min. to form a homogeneous system. Then, degassing it in vacuum drying oven to remove residual solvent and bubbles to obtain nano-PDA@GO-TiO$_2$ composites waterborne epoxy anticorrosive coating (PGT-2%/WEP). Finally, the prepared coating layer was evenly coated on the surface of the treated Q235 steel test piece, and the coating thickness was controlled between 100 and 120 μm with the help of film applicator, and cured at room temperature for 72 h. On the purpose of comparson the effect of different content of nano-PDA@GO-TiO$_2$ composites for the coating properties, The different content of nano-PDA@GO-TiO$_2$ composites waterborne epoxy coatings were prepared and named as WEP, PGT-0.5%/WEP, PGT-1%/WEP, GPT-2%/WEP, and GPT-3%/WEP in a similar way.

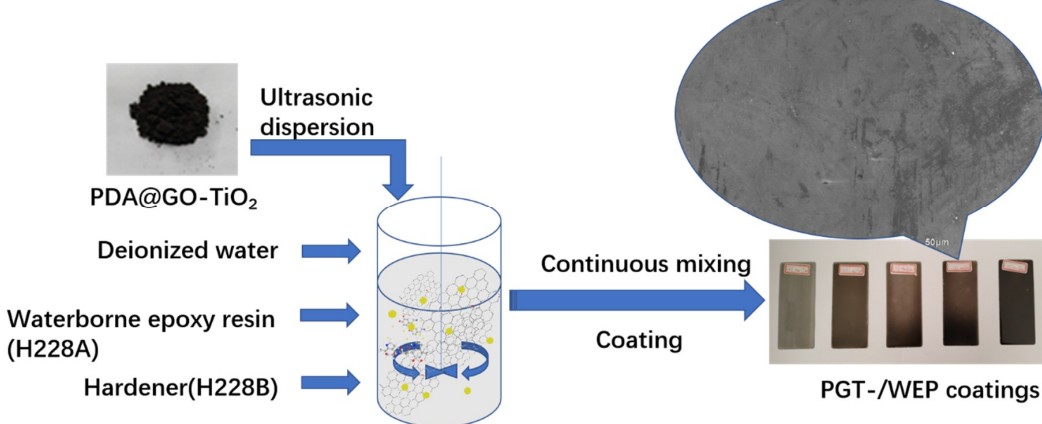

**Figure 2.** Chart of preparation of composite coating.

### 2.4. Characterization and Testing

### 2.4.1. Characterization

The chemical constitutions of nano-PDA@GO-TiO$_2$ composites, graphene oxide (GO), and nano-TiO$_2$ were characterized by FT-IR. The spectrums were recorded on Fourier transform infrared spectrometer (FT-IR, Nicolet IS5, Thermo Fisher, Madison, WI, USA) over the wave number range of 4000–400 cm$^{-1}$. The testing samples were prepared by Potassium bromide tableting. The nano-PDA@GO-TiO$_2$ composites, graphene oxide (GO) and nano-TiO$_2$ were also analyzed by X-ray photoelectron spectroscopy (XPS, Thermo ESCALAB 250XI, Thermo Fisher Scientific, Madison, WI, USA) to determine the composition. The crystal structure of the nano-PDA@GO-TiO$_2$ composites, graphene oxide (GO), and nano-TiO$_2$ were examined by X-ray diffraction (XRD, D/MAX2500, Rigaku, Tokyo, Japan) with "Cu K$\alpha$" radiation (1.540A) source at θ between 5° and 80°, the diffraction angle(2θ) was scanned at a rate of 5°/min. The surface structure and morphology of the coating were observed by scanning electron microscopy (SEM, SUPRA 55, Carl Zeiss AG, Jena, Germany). Raman characterization was operated on an in Via Reflex confocal Laser Micro Raman spectrometer (Raman, Lab RAM HR Evolution, HORIBA, Market (China) Trading Co., Ltd. Shanghai, China), excitation source: 532 nm Ar+ laser and power of 25%. The microstructure and morphology of GO and nano-PDA@GO-TiO$_2$ composites were estimated by transmission electron microscopy (TEM, JEOL 2100, Japanese electronics, Tokyo, Japan).

### 2.4.2. Performance Testing of Coatings

Physical property of coatings testing: The coatings was tested by the electric paint adhesion tester (QFZ Shanghai Leao Experimental Instrument Co., Ltd. ShangHai, China). The impact resistance was tested by the paint film impactor (QCJ-120, Shanghai Leao Experimental Instrument Co., Ltd. ShangHai, China). The hardness of the coating was tested by a pencil hardness tester (QHQ-A, Shanghai Leao Experimental Instrument Co., Ltd. ShangHai, China).

Electrochemical performance of coatings testing: Electrochemical performances of the composite coatings were investigated while using an electrochemical workstation of the three-electrode cell (CHI920D scanning electrochemical microscope workstation, Shanghai Chen hua Instrument Company, ShangHai, China). Platinum was wire electrode acted as an auxiliary electrode, and a silver chloride electrode containing saturated potassium chloride solution was served as a reference electrode, Coated with a composite coating Q235 test strip (1 × 1 cm$^2$) as a working electrode, 3.5 wt.% NaCl solution was used as the electrolyte. The working electrode was immersed in a 3.5% sodium chloride solution, and the electrochemical impedance spectroscopy spectrum and the Tafel curve of the coating were tested after the system was stabilized. The electrochemical AC impedance spectrum test frequency range is 10$^{-2}$~10$^5$ Hz. The scan frequency of the Tafel curve test is 0.01 V/s, and the scan potential is open circuit voltage ±0.25 V. Electrochemical testing of all the samples was repeated three times to ensure data repeatability.

## 3. Results and Discussion

### 3.1. FT-IR Spectroscopy

FT-IR spectra of TiO$_2$, GO and nano-PDA@GO-TiO$_2$ composites are shown in Figure 3. As for GO, many clear strong absorption peaks at 3415, 1727, 1617, 1405, and 1060 cm$^{-1}$ are ascribed to the stretching vibrations of water O-H stretching, carboxylates or ketones C=O stretching, water O-H bending and C=C stretching, epoxide C−O−C or phenolic C−O−H stretching, and C−O stretching, respectively [29,30]. As to the spectrum of TiO$_2$, the absorption around 575 cm$^{-1}$ and 1115 cm$^{-1}$ are assigned to the vibration of Ti–O–Ti and Ti–O bonds in TiO$_2$ [31,32]. Compared with pure GO, after combination with PDA and TiO$_2$, the peak of C=O stretching vibrations at 1727 cm$^{-1}$, almost disappeared in nano-PDA@GO-TiO$_2$ composites, there are mainly attributed to the removal of oxygen

functionalities from GO as a result of the reduction effect of PDA. In addition, the new peaks that appear at 1055 cm$^{-1}$ and 1504 cm$^{-1}$ are corresponding to the of C–N stretching and aromatic C=C stretching/ N–H bending that originated from PDA, respectively [30]. In the spectrum of PDA@GO-TiO$_2$, the broad band between 400–1000 cm$^{-1}$ stretching in a TiO$_2$-based composites [31]. In addition, after the addition of dopamine, the color of the GO suspension changed from yellowish brown to black, that also implying the reduction of GO to grapheme [33,34]. These results support that the preparation of nano-PDA@GO-TiO$_2$ composites with our process is successful.

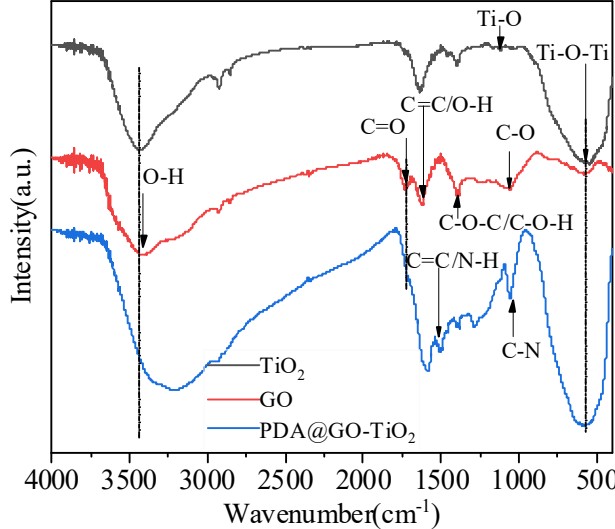

**Figure 3.** Fourier transform infrared spectrum of nano-PDA@GO-TiO$_2$ composites, graphene oxide (GO), and nano-TiO$_2$.

### 3.2. XRD Patterns

XRD patterns of GO, nano-TiO$_2$ and nano-PDA@GO-TiO$_2$ composites are shown in Figure 4. The specific XRD of TiO$_2$ is characterized by five peaks that are positioned at the 2θ values of 25.31°, 38.01°, 48.02°, 54.45°, and 62.59°, which correspond to the(101),(112),(200),(105), and(204) lattice Plane of anatase phase of TiO$_2$ (JCPDS card no. 21-1272), respectively [35]. The GO shows a broad peak at 2θ = 12.32°, which corresponds to the characteristic diffraction peak of GO [36]. In addition, Aweak derivative peak appeared around 26°, which should be attributed to the crystallization peak of residual graphite oxide [37]. However, after modification with dopamine, A new broad diffraction peak at 23.5° can be observed for the resultant nano-PDA@GO-TiO$_2$ composites, which was due to the stacking of the benzene rings of PDA molecules, demonstrates that the DA has successfully self-polymerized on the GO [38]. The reduction of GO by DA modified is also established by the appearance of broad peak observed in the region (22°–38°) and the disappearance of the peak at 12.32° in nano-PDA@GO-TiO$_2$ composites of XRD [34,39]. At the same time, the spectrum of XRD for nano-PDA@GO-TiO$_2$ composites was similar to nano-TiO$_2$. This indicates that nano-TiO$_2$ is successfully loaded on the surface of GO to form nano-PDA@GO-TiO$_2$ composites.

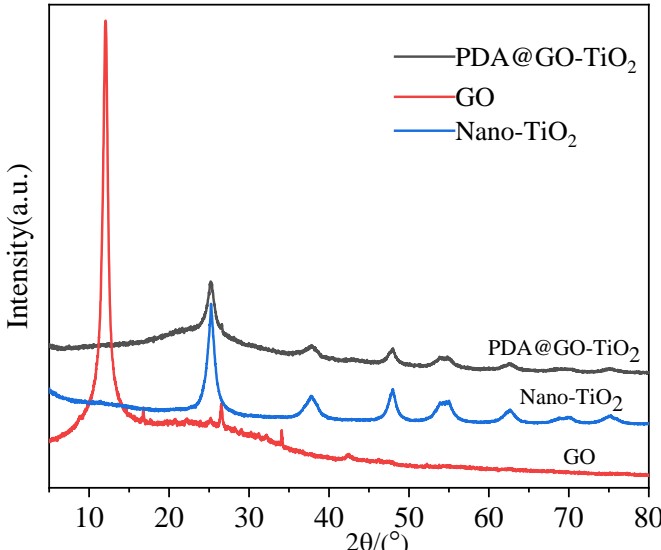

**Figure 4.** X-ray diffraction (XRD) patterns of Nano-TiO$_2$, GO, nano-PDA@GO-TiO$_2$ composites.

### 3.3. Raman Spectra Analysis

Raman spectra for GO and nano-PDA@GO-TiO$_2$ composites are shown in Figure 5. It can be seen from Figure 5 that GO exhibits the D band at 1356 cm$^{-1}$ (breathing mode of A$_{1g}$ symmetry) and G band at 1587 cm$^{-1}$ (E$_{2g}$ symmetry of sp$^2$ carbon atoms), respectively [40]. For PDA@GO-TiO$_2$, peaks are clearly observed at 1305 and 1544 cm$^{-1}$, which can be assigned to the D and G bands. The peaks at 106 cm$^{-1}$, 344 cm$^{-1}$, 450 cm$^{-1}$, and 582 cm$^{-1}$ in the PDA@GO-TiO$_2$ Raman spectrum should be assigned to the E$_{g(1)}$, B$_{1g(1)}$, A$_{1g}$ + B$_{1g(2)}$, and E$_{g(2)}$ mode of TiO$_2$, which indicates the successful incorporation of TiO$_2$ into rGO [41,42]. It is also found that, after function with PDA, shows red shifting of the D band peak from 1356 cm$^{-1}$ to 1305 cm$^{-1}$ shows and the G band peak shifts from 1587 cm$^{-1}$ to 1544 cm$^{-1}$. Red shifting in G and D-bands of PDA@GO-TiO$_2$ suggests that the reduction of dopamine lead to removal of oxygenated functionalities from GO surface [40,43,44]. In addition, the intensity ratio of the D and G bands, ID/IG, increases from 0.89 for GO to 1.04 for PDA@GO-TiO$_2$. The results show that the grafting of PDA on graphene and the loading of TiO$_2$ produce more defects and reduces the average size of the sp2 domain [45].

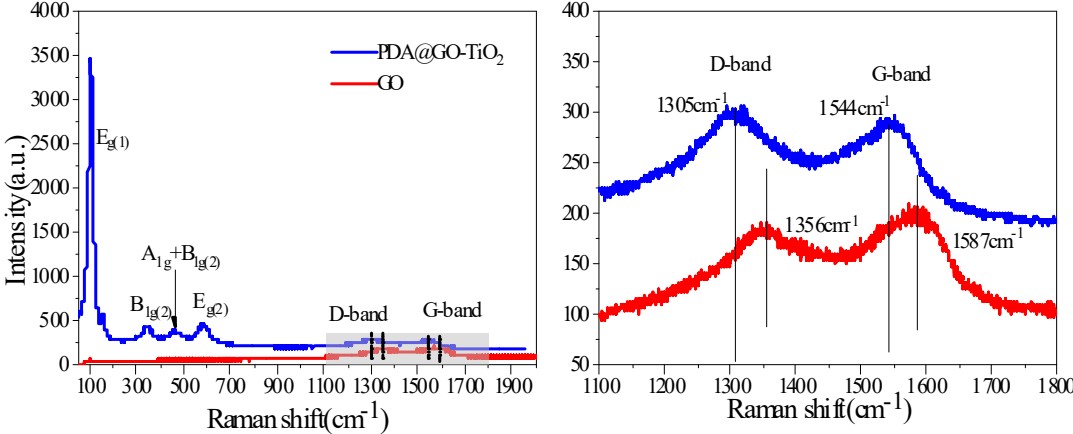

**Figure 5.** Raman spectra of GO and nano-PDA@GO-TiO$_2$ composites.

### 3.4. XPS Analysis

Considering the detection limit of FT-IR measurement, the X-ray photoelectron spectroscopy of samples were employed to evaluate the detailed chemical bonds formed on the surfaces of GO

before and after its functionalization with PDA and $TiO_2$. Figure 6 gives the XPS spectra of GO and nano-PDA@GO-$TiO_2$ composites. As the spectrum of GO (Figure 6a) and PDA@GO-$TiO_2$ (Figure 6b) shown that the atomic percentage of C and O in GO was found to be 67.9% (C) and 32.1% (O), whereas it was 70.2% (C) and 23.2% (O) in nano-PDA@GO-$TiO_2$ composites. The significant decrease in the percent of O confirms the removal of oxygen containing functionalities. Figure 6c,d shows the C 1 s spectra of GO and PDA@GO-$TiO_2$, respectively. The C 1 s spectrum of GO (Figure 6c) can be curved into four peak components with binding energies at about 284.63, 286.73, 287.80, and 289.03 eV, attributable to the C–C/C=C, C–O, C=O, and O–C=O species, respectively. After reduction and modification by DA, the peaks of the C–O groups disappear, which indicates that most of the epoxide and carboxyl functional groups are removed and the GO is reduced to PDA@GO [46,47]. More over the new strong peaks at 285.23 eV assigned to the C–N groups increase in the PDA@GO-$TiO_2$, as shown in Figure 6d, which may be attributed to the catechol/amino groups of the adhered PDA. on the other hand, The deconvolution of the N1s peak at 400 eV (Figure 6e) can be deconvoluted into three peaks at 399.7 ($NH_2$ and $\underline{N}$=N=N), 401.6 (NH−C=O), and 403.5 eV(N=$\underline{N}$=N) consistent with the chemical composition of the dopamine derivative, which indicated that the incorporation of azide-terminated dopamine in the reduced graphene nanosheets [48]. Compared with the GO, the high resolution XPS scan of the Ti2p region demonstrates the presence of the PDA@GO-$TiO_2$ (Figure 6f). The Ti2p3/2 and 2p1/2 peaks occur at 458.6 and 464.3 eV, respectively [49].

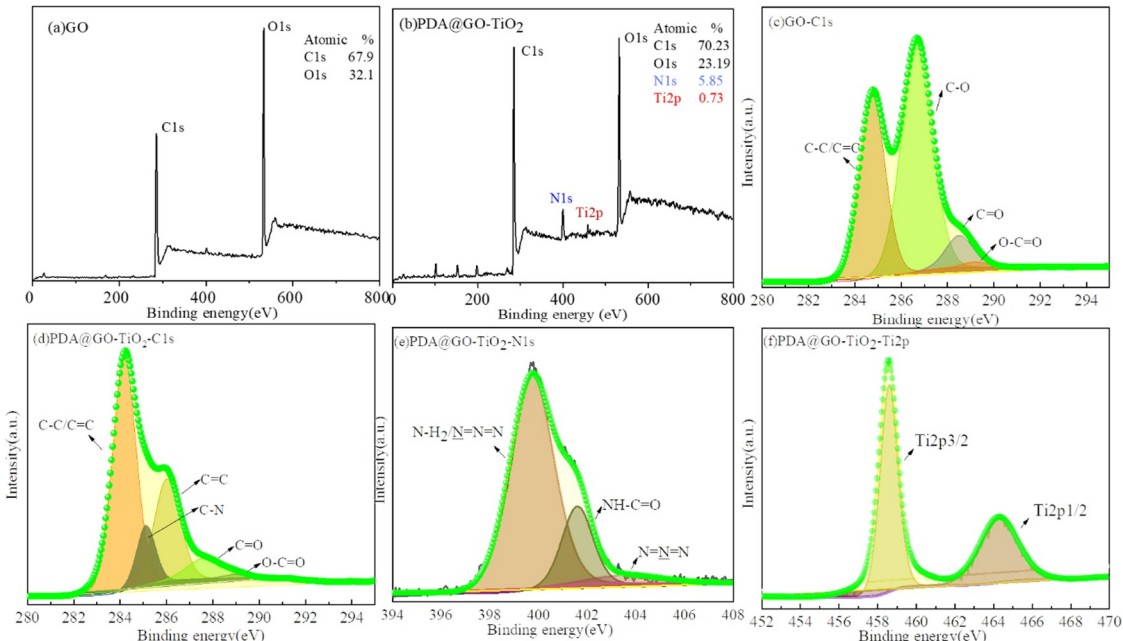

**Figure 6.** X-ray photoelectron spectroscopy (XPS) spectra of (**a**) GO, (**b**) PDA@GO-$TiO_2$, (**c**) GO/C1s, (**d**) PDA@GO-$TiO_2$/C1s, (**e**) PDA@GO-$TiO_2$/N1s, and (**f**) PDA@GO-$TiO_2$/Ti2p.

### 3.5. TEM Analysis

The morphologies of GO and nano-PDA@GO-$TiO_2$ composites were observed by TEM, as shown in Figure 7a–e. It can be seen from Figure 7a,b that GO that exhibits a typical translucent sheet structure with curls and wrinkles in the middle portion, which may be due to a deformation during peeling or a superposition effect of GO sheets. As shown in Figure 7c,d, the PDA@GO layer structure can effectively prevent the aggregation of $TiO_2$ nanoparticles and the decoration of $TiO_2$ nanoparticles has good dispersibility on the surface of PDA@GO. From Figure 7e, it can be seen that the nano-$TiO_2$ particles on the surface of PDA@GO have good crystallinity and lattice spacing, the relatively obvious lattice spacing of 0.352 nm, which are in good agreement with the nano-$TiO_2$ particles in XRD at $2\theta = 25.2°$ (101) crystal plane. In addition, the sedimentation test of GO and nano-PDA@GO-$TiO_2$ composites

in deionized water, as shown in Figure 7a,c, which presents the dispersing ability of the GO and nano-PDA@GO-TiO$_2$ composites in deionized water. In this test, GO and PDA@GO-TiO$_2$ are dispersed in deionized water with ultrasonication for 45 min and then the dispersions are storage without any disturbance for same times. As shown in Figure 7a,c, GO has deposited into the bottom of the bottle completely after 3 h of storage, while there is no obvious stratification for the nano-PDA@GO-TiO$_2$ composites solution at all the time, indicating that the dispersing ability of the nano-PDA@GO-TiO$_2$ composites in deionized water is significantly. This may be attributed to that covalent grafting exists between the active sites of the GO and DA molecules, which promotingthe GO is reduced to Graphene, and improving the dispersing ability of the nano-PDA@GO-TiO$_2$ composites in deionized water.

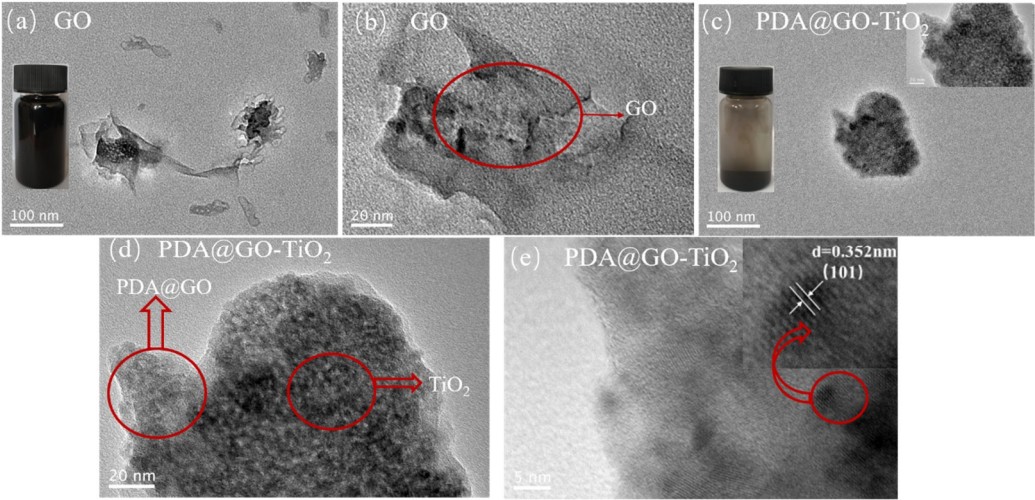

**Figure 7.** Transmission electron microscopy (TEM) images of GO (**a**,**b**) and nano-PDA@GO-TiO$_2$ composites (**c**–**e**).

### 3.6. SEM Analysis

The SEM was used to further observe the microscopic morphology of the composite coatings. Figure 8a–e and a'–e' show the surface and cross section microscopic structure of WEP, PGT-0.5%/WEP, PGT-1%/WEP, PGT-2%/WEP, and PGT-3%/WEP, respectively. It can be seen from Figure 8a that the surface of the WEP coating is rough and there are a large number of micropores, the WEP coating cross section (Figure 8a') exhibits a relatively smooth river-shaped fracture surface, and the fracture surface has many microporous structures, which may be due to the micropores remaining on the surface after the resin curing. With the nano PDA@GO-TiO$_2$ composites added to the surface of the coating, the micropores gradually decrease among them, the surface of PGT-2%/WEP coating (Figure 8d) has a single microstructure and high regularity. In addition, it can be seen from the cross-section of PGT-2%/WEP coating (Figure 8d') that nano-PDA-GO-TiO$_2$ composite exhibits good compatibility with resin. This is because DA forms a polydopamine (PDA) layer on the surface of GO by self-polymerization and deposition, which improves the compatibility of GO with waterborne epoxy resin and enhances the interfacial adhesion between GO and epoxy resin. However, PGT-3%/WEP coating (Figure 8e) has defects (microcracks) in the coating, which may be due to the accumulation of excess nano-PDA @GO-TiO$_2$ composites outside the voids of the resin space structure, which cannot be completely covered by the resin. This can be confirmed by the cross-section of the PGT-3%/WEP coating (Figure 8e').

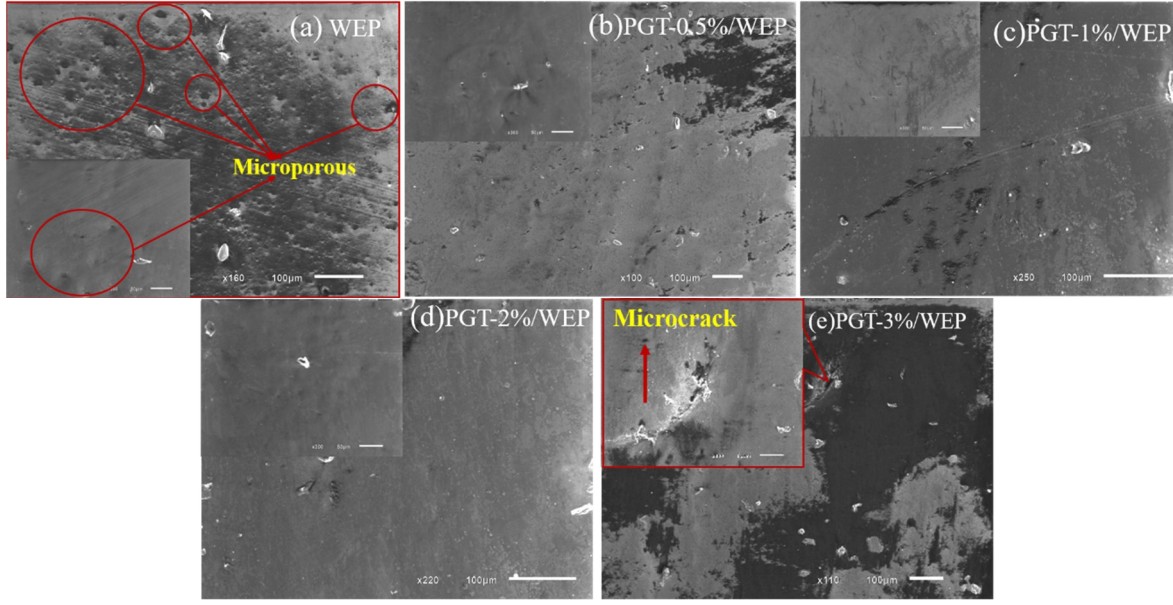

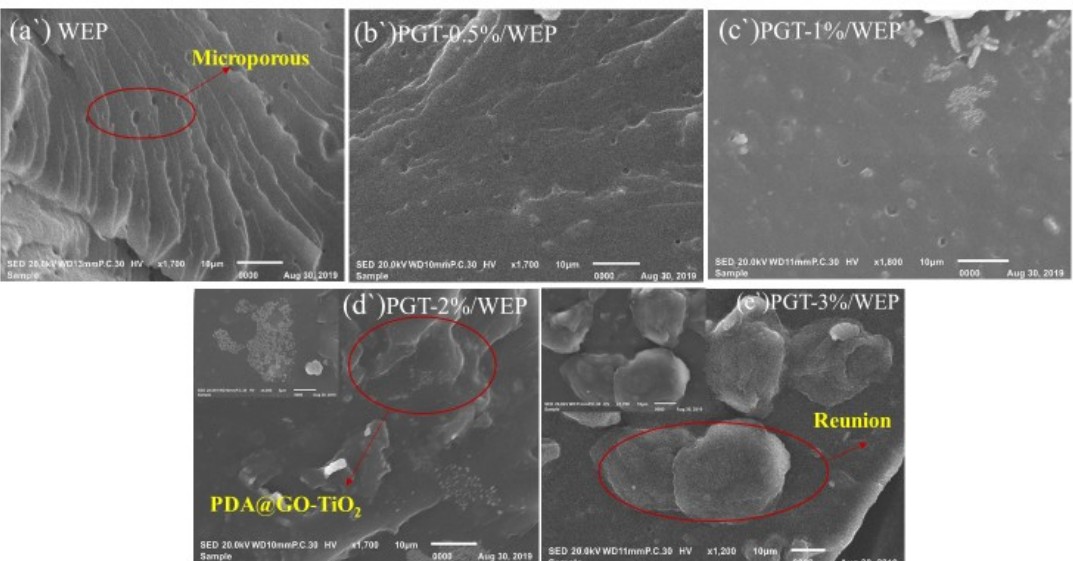

**Figure 8.** Scanning electron microscopy (SEM) surface image of (**a**) waterborne epoxy resin coating (WEP), (**b**) (PGT)-0.5%/ WEP, (**c**) PGT-1%/ WEP, (**d**) PGT-2%/ WEP and (**e**) PGT-3%/ WEP coating. SEM cross-sectional image of (**a'**) WEP, (**b'**) PGT-0.5%/ WEP, (**c'**) PGT-1%/ WEP, (**d'**) PGT-2%/ WEP and (**e'**) PGT-3%/ WEP coating.

### 3.7. Mechanical Property Characterization

Evaluation of the mechanical properties of the WEP and PGT/WEP coatings by impact resistance, pencil hardness and adhesion are shown in Figure 9a–c. The impact resistance of the WEP coating is 45 cm (Figure 9a). The pencil hardness is 2H (Figure 9b) and the adhesion rating of 2 (Figure 9c). In contrast, The PGT/WEP coatings containing nano PDA@GO-TiO$_2$ composites have high impact resistance, pencil hardness, and adhesion. Among, the pencil hardness of the PGT-2%/WEP coating up to 5H, the impact resistance up to 60 cm and the adhesion rating of 1. Since nano-filler (nano-PDA@GO-TiO$_2$ composite) is completely coated by resin, it plays a reinforcing role in the spatial structure of resin. Filling the voids of the polymer segment itself and improving the Interfacial interaction of coating-metal. When the coating is impacted, the nanofiller absorbs part of the impact

energy to improve the mechanical properties, such as adhesion and impact resistance of the coating. So, the PGT-2%/WEP coating exhibiting excellent physical and mechanical properties. However, it is obvious from Figure 9a–c, the physical functions such as impact resistance and pencil hardness of the PGT-3%/WEP coating has a tend to reduce. As a result of adding excessive amounts of nano-filler(nano-PDA-GO-TiO$_2$) in coating, the nano-filler cannot be completely coated by the resin, and part of PDA-GO-TiO$_2$ is concentrated outside the void of the resin space structure, resulting in the bond between the coating and the substrate is affected, and the hardness of the coating, performance, such as impact resistance, is reduced. The mechanism analysis is as follows: The first, GO modified by dopamine is successfully reduced and stripped. Dopamine anchored between GO lamellar structures, which can effectively inhibiting the agglomeration between the GO sheets, and is beneficial to the encapsulation of the GO sheet by the resin, further improving the dispersing ability of the nano-PDA@GO-TiO$_2$ composites in resin matrix. On the other hand, depositing nano-TiO$_2$ on the surface of graphene oxide (GO) sheets can prevent graphene oxide (GO) nano sheets from restacking. at the same time, the presence of TiO$_2$ nanoparticles in epoxy pores inducing the healing effect by sliding and filling pinholes, interstitial spaces of coating micro cracks regions, further to improve the hemechanical properties of PGT/WEP coating. The third, DA, forms a polydopamine layer by self-polymerization and deposits on the surface of GO. There are rich in active energy groups (such as burlyandum and midamin) of DA and PDA, it not only interact with the oxygen-containing metastases on the surface of GO to produce rGO tablets, but also reacts with the epoxy base of the epoxy resin, enhancing the binding force between the interface between nano-PDA@GO-TiO$_2$ composites and epoxy resin, and improving the mechanical properties of the coating.

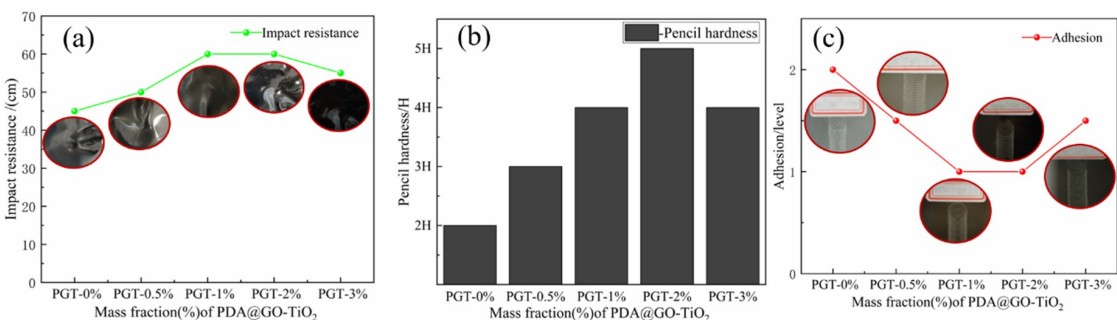

**Figure 9.** Impact resistance (**a**), Pencil hardness (**b**), Adhesion (**c**) of PGT/WEP coatings with different contents of nano-PDA@GO-TiO$_2$ composites.

The physico-mechanical properties that figure out the Impact resistance, hardness and adhesion of the coating were estimated and tested according to following standards:

- Adhesion refers to GB/T 1720—1979(89), Coating adhesion test method (circle method);
- Impact resistance refers to GB/T 1732—1993, Paint film impact resistance test (Impact test); and,
- Hardness refers to GB/T 6739—2006, Pencil method for measuring film hardness (Pencil hardness);

### 3.8. Electrochemical Characterization

The corrosion kinetics of the PGT/WEP coatings with different PDA@GO-TiO$_2$ content on the surface of the metal body were studied by polarization curves. The polarisation curves of the four nano-PDA@GO-TiO$_2$ composites PGT/WEP coatings and pure WEP coating after immersion for 7 days are provided in Figure 10, and the parameters extracted from polarization curves after Tafel fitting as shown in Table 1. In usually, corrosion potential is used as a thermodynamic parameter to reflect the thermodynamic state of the coating. Corrosion electro-corrosion potential is used as a thermodynamic parameter to reflect the thermodynamic state of the coating. The more positive the corrosion potential (E$_{coor}$) represents the better corrosion resistance of the coating. The better the

corrosion current is used as the kinetic parameter, the dynamic state of the coating. The smaller the corrosion current density ($I_{coor}$) represents the slower the corrosion rate of the coating [50]. From Figure 10, the WEP coating of self-corrosion potential ($E_{coor}$) was−0.656 V, with the increased of nano-PDA@GO-TiO$_2$ composites, the self-corrosion potential of PGT-1%/WEP coating was −0.514 V and the PGT-2%/WEP coating was −0.437 V. At the same time, the self-corrosion current ($i_{corr}$) of the WEP, PGT-0.5%WEP, and PGT-1%/WEP were $0.314 \times 10^{-6}$ A·cm$^{-2}$, $0.238 \times 10^{-}$ A·cm$^{-27}$, and $0.725 \times 10^{-8}$ A·cm$^{-2}$, respectively. The self-corrosion current density of PGT-2%/WEP coating was as low as $0.418 \times 10^{-9}$ A·cm $^{-2}$. However, a clear shift towards negative potentials appear with the further addition of nano-PDA@GO-TiO$_2$ composites, and the $i_{corr}$ increased to $0.479 \times 10^{-8}$ A·cm$^{-2}$ for PGT-3%/WEP coating. So, the corrosion protection performance against corrosion could be established in the following order: PGT-2%/WEP > PGT-3%/WEP > PGT-1%/WEP > PGT-0.5%/WEP > pure WEP coating. In addition, the PGT-2%/WEP coating demonstrated a reduction of two or three orders of magnitude in corrosion rate as compared to WEP coating.

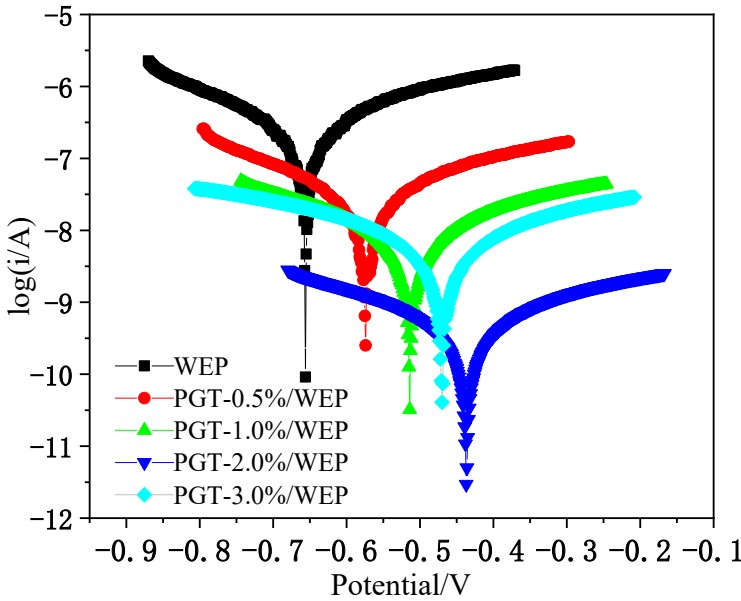

**Figure 10.** Polarization curves of PGT/WEP coatings immersed in 3.5 wt.% NaCl aqueous solution for 7 days.

**Table 1.** Electrochemical corrosion study of WEP coatings with nano-PDA@GO-TiO$_2$ composites in 3.5% NaCl corrosive conditions.

| Sample | $E_{coor}$/v | $i_{coor}$/A·cm$^{-2}$ | $\beta_a$/mV dec$^{-1}$ | $\beta_c$/mV dec$^{-1}$ | $R_p$ (KΩ·cm$^2$)/E6 |
|---|---|---|---|---|---|
| WEP | −0.656 | $0.314 \times 10^{-6}$ | 206.79 | −209.29 | 3.327 |
| PGT-0.5%/WEP | −0.573 | $0.238 \times 10^{-7}$ | 192.49 | −203.79 | 46.098 |
| PGT-1%/WEP | −0.514 | $0.725 \times 10^{-8}$ | 203.58 | −205.17 | 146.98 |
| PGT-2%/WEP | −0.437 | $0.418 \times 10^{-9}$ | 209.55 | −206.01 | 2539.43 |
| PGT-3%/WEP | −0.459 | $0.479 \times 10^{-8}$ | 206.82 | −201.12 | 222.57 |

Electrochemical impedance spectrometry (EIS) to further evaluate the anticorrosive properties of PGT/WEP coatings with different nano-PDA@GO-TiO$_2$ composites content at different immersion times, as illustrated in Figure 11a–f. In general, the corresponding impedance module that sat under the lowest frequency is utilized in Bode impedance plots. $|Z|_{0.01}$ could be used as a semi-quantitative indicator of the corrosion resistance of the coating [51]. As shown in Figure 10a–f, for the pure WEP coating, the impedance modulus polts at $|Z|_{0.01}$ was $1.39 \times 10^4$ Ω·cm$^2$ at the 1 day of immersion of 3.5% NaCl. However, the value decreased to $1.01 \times 10^3$Ω·cm$^2$ after 15 days of immersion, then reduced

to $6.37 \times 10^2 \Omega \cdot cm^2$ after 30 days of continuous immersion, indicating the diffusion of Corrosive ion into the WEP coatings and resulting in loss of protective ability of the coating. In comparison, PGT-0.5%/WEP, PGT-1%/WEP, GPT-2%/WEP, GPT-3%/WEP coatings, the initial impedance modulus at $|Z|_{0.01}$ were $5.31 \times 10^4 \ \Omega \cdot cm^2$, $6.91 \times 10^4 \ \Omega \cdot cm^2$, $8.62 \times 10^4 \ \Omega \cdot cm^2$, and $6.67 \times 10^4 \ \Omega \cdot cm^2$, and these values decreased to $5.23 \times 10^3 \ \Omega \cdot cm^2$, $7.34 \times 10^3 \ \Omega \cdot cm^2$, $1.02 \times 10^4 \ \Omega \cdot cm^2$, $6.53 \times 10^3 \Omega \cdot cm^2$ after immersion of 30 days, which is one to two orders of magnitude higher than the pure WEP coating during the whole immersion process. This is mainly due to the good dispersibility of the nano-PDA@GO-TiO$_2$ composites in the aqueous epoxy matrix. Obviously, the GPT-2%/WEP coating exhibited the highest impedance modulus than those of other coatings during the whole immersion process, which suggesting that the coating possessed the best anticorrosion performance.

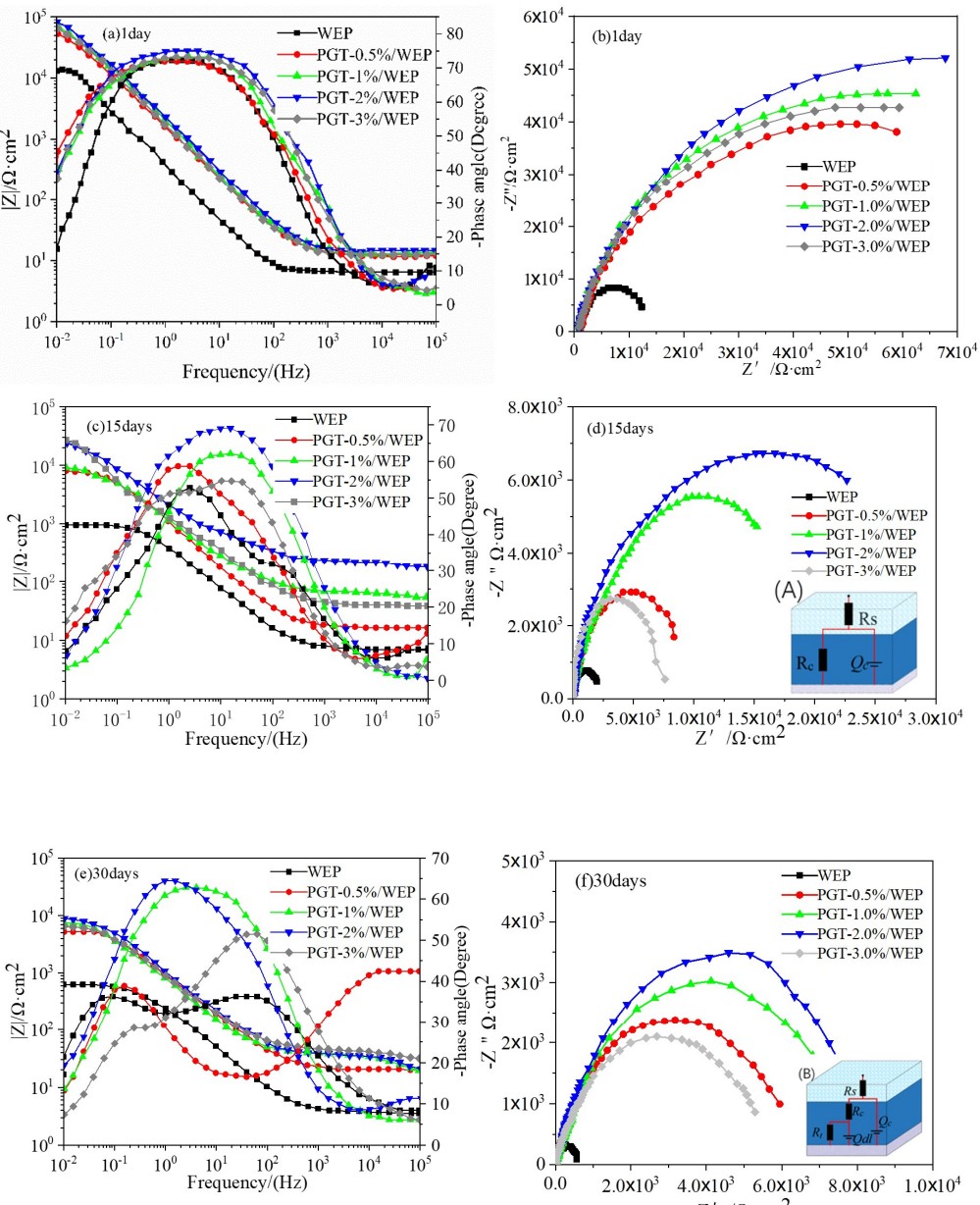

**Figure 11.** Bode impedance diagram, Bode phase diagram and Nyquist diagram of PGT/WEP coating soaked for 1 day, 15 days, and 30 days, respectively. (A), (B) The equivalent circuits.

In order to further analyze the protective properties of the PGT/WEP coatings for the substrate the equivalent circuit was used to fit the EIS curve, as shown in Figure 11d–f. In the circuit, Rs, Rc, and Rct

represented the electrolyte resistance, coating resistance and charge transfer resistance, respectively. Qc and Qdl represented the coating capacitance and the electric double layer (steel/colution) capacitance. The corrosion process of the coating is mainly divided into two stages: the period in which the corrosive medium has not penetrated to reach the coating/base metal interface is called the first stage, and the A model is used in this stage. As the corrosive medium penetrates the interface of the coating/substrate metal and forms a corroded microbattery, the impedance spectrum that was measured at this time has two times constants, which is called the second stage, and the B model is used in this stage [52]. From Figure 11c, it is clear that the pure WEP coating begins to appear the second time constants during immersion for 15 days. At the same time, the GPT-3%/WEP coating shows a similar situation. It is worth nothing that the GPT-2%/WEP coatings always maintained the highest Rc and phase angle values, indicating that the GPT-2%/WEP coating has a strong barrier effect. As shown in from the Nyquist diagram of Figure 11a–f, that the capacitance arc of the pure WEP coating is much smaller than the capacitance arc of the GPT/WEP coating, and the GPT-2%/WEP coating has a higher capacitance arc. After 15 days of immersion for 30 days, as in Figure 10f, the GPT-2%/WEP coating still exhibited a larger capacitance arc than the other coatings. Because of the nano PDA@GO-TiO$_2$ composites can block the micropores in the coating to form a shielding layer. Effectively blocking the infiltration of corrosive media such as O$_2$, Cl$^-$, and H$_2$O, thus greatly reducing the corrosion rate and significantly improving the long-term anticorrosive properties of water borne epoxy coatings. On the contrary, the capacitance arc of the GPT-3%/WEP coating is significantly smaller than that of other GPT/WEP coatings, which may be caused by excessive aggregation or uneven dispersion of the nano PDA @ GO-TiO$_2$ composite in the resin matrix, resulting in weakening of the interface between the nano PDA@GO-TiO$_2$ composites and the resin and forming cracks around the aggregation region, leading to the decrease of anticorrosive properties.

### 3.9. Anticorrosion Mechanism Analysis

Based on the above discussion, the nano-PDA@GO-TiO$_2$ composites can significantly improve the corrosion resistance of waterborne epoxy coatings. Its anti-corrosion mechanism is shown in Figure 12. First, DA enhances the interface binding force between PDA-GO-TiO$_2$ and epoxy resin by self-polymerizing and depositing on the surface of GO to form a polydopamine (PDA) layer, and the active metabody (beramand and midamin) of the DA and PDA can react with the epoxy substrate of the epoxy resin. In addition, PDA and the oxygen-containing functional group interaction on the surface of GO, so that graphene oxide reduction peeling and anchor between its sheet structure, increase the surface area and enhance the compatibility of the GO in the water-based epoxy resin dispersion. The uniform dispersion of nano-PDA-GO-TiO$_2$ composites in water-based epoxy resin layer overlay, which forming a dense physical barrier layer. The GO and TiO$_2$ have synergies, by the filling resin cured after the micro-cracks and pores to achieve healing effect, and in the coating-metal interface caused by strong shielding and locking effects. Making the diffusion penetration path of the corrosive medium more complex, which effectively improves the anti-penetration ability of the coatings.

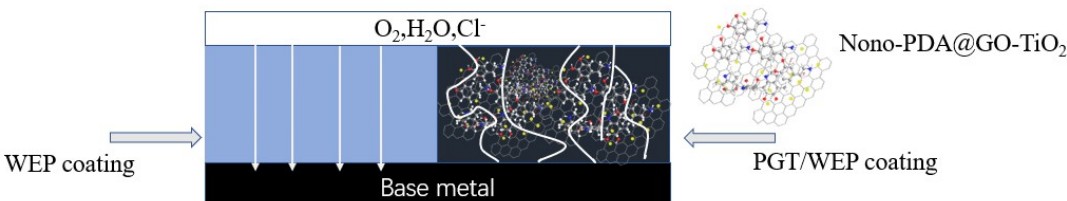

**Figure 12.** Diagram of the path of corrosive media (O$_2$, H$_2$O, Cl) through pure waterborne epoxy coating (WEP) and PGT/WEP coating.

## 4. Conclusions

In summary, nano-PDA@GO-TiO$_2$ composites are successfully prepared by a facile solution method, the nano-PDA@GO-TiO$_2$ composites as the nanofiller for reinforcing physical properties and anticorrosion performance of waterborne epoxy coatings, and the anticorrosion mechanisms are investigated in detail. Through the basic physical properties (hardness, impact resistance, adhesion) and electrochemical tests, show that the addition of nano-PDA@GO-TiO$_2$ composite materials can improve the hardness, adhesion, and impact resistance of waterborne epoxy resin. The hardness of PGT-2%/WEP coating reaches 5H, and the adhesion is grade 1. The density I$_{coor}$ is as low as $0.418 \times 10^{-9}$ A.cm$^{-2}$ and the impedance value is maximized. In addition, PGT-2%/WEP coating exhibits excellent mechanical properties and corrosion resistance compared to other coatings. The enhancement of physical properties and anticorrosion capability could be ascribed to the strong interfacial bonding between GO and waterborne epoxy after modification by PDA, which uniformly dispersed in the resin to form a dense physical barrier layer, and nano-TiO$_2$ particles deposited on the surface of GO in resin matrix as nanofiller, it fills the pinhole interstitial crosslinked spaces and other coating artifacts (micro cracks and voids) regions not only provide strength to the coating material, but also enhance the corrosion protective efficiency of the coatings. Having a strong barrier effect on corrosive media, such as O$_2$, H$_2$O, and Cl$^-$. This study shows that anticorrosion and physical properties of PGT/WEP coatings can be enhanced with the incorporation of nano-PDA@GO-TiO$_2$ composites.

**Author Contributions:** J.Z. initiated the idea. J.Z. and J.C. performed the experiments and took the photographs in this article. J.Z. wrote the paper and analyzed the data. H.B. collected and analyzed the results. S.W., Y.R., B.L. and S.Z. provided helpful ssion in order to improve the manuscript. S.W. and J.Z. supervised and gave guidance on the experimental design, analyzing data, and writing the manuscript.

**Funding:** The financial support from National Nature Science Foundation of China under the contract (51574045) and Jiangsu Province Graduate Research and Practice Innovation Project (SJCX18_0967).

**Conflicts of Interest:** The authors declare no conflict of interest.

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
