# Peer review of "Polydopamine Modified Graphene Oxide-TiO2 Nanofiller for Reinforcing Physical Properties and Anticorrosion Performance of Waterborne Epoxy Coatings"

_applsci, doi:10.3390/app9183760_

Round 1

Reviewer 1 Report

Polydopamine Modified Graphene Oxide-TiO2 Nanofiller for Reinforcing Physical Properties and Anticorrosion Performance of Waterborne Epoxy Coatings.

Since the first time submission this manuscript has been  improved. The authors have made an attempt to provide a bit deeper explanation of the results and the introduction chapter has been improved.

However, in the current state it is difficult to recommend the manuscript to be accepted since the English language is not appropriate for a scientific paper. Also, too many inaccuracies in the manuscript still exist. In overall, it is difficult to follow the manuscript information due to the language quality, just some examples from introduction:

Line 18: Results demonstrate that the introduction of dopamine to functionalize the GO could selfpolymerized polydopamine (PDA) on the surfaces of the GO and endow abundant chemical groups reduce the GO.  Should be, for instance, could self polymerize

Line 32 : It is a known fact that damage to petrochemical equipment, oil and gas pipelines, machinery and other infrastructure caused by metal corrosion has caused serious economic losses, resource waste and environmental pollution. Difficult to understand

Line 37: Epoxy resin coatings has widely been. Should be, - have

Line 39: as the most effective method to inhibit corrosion of the metal

substrate of surface [2, 3]. What do you mean the metal substrate of surface?

Line 50: It shows more or less permeability to corrosive

substances (ie H2O, O2 and Cl-). Probably high permeability to

“many researchers have begun to apply nanotechnology

to waterborne epoxy coatings to. “

"In order to further improve the compatibility of GO in polymer composites. Which make interfacial adhesion between the GO and polymer composites is improved."

And many more examples of English low quality and many mistakes and inaccuracies.

Materials and methods:

Line 144:  “the coating thickness was controlled between 100 and 120μm”. How did you measure it and how accurate these values?

Line 167 is the repeat.

Results:

The peak shift in the Raman spectra for the nanocomposite can also indicate that there is a pressure inserted from the “matrix”. I guess a similar result can be observed even for PDA:GO or any other polymer system. No possible laser heat damage is discussed.

Line 293: “the micropores gradually decrease and the surface of the PGT-2%/WEP coating (Fig. 8d) is hardly to see the micropores.” I do not think authors have enough support in the presented SEM figures to claim it. Clearly, quite big pores, i.e. several um, are presented in all figures. Also, how representative these few images are? Are these figures were measured in SE or BSE? There is an interesting contrast for the GO/TiO2 containing coatings, which can suggest that there are big aggregates in the coatings. Thus, entire measured properties can significantly vary with the measured areas.

Mechanical properties. It is important to show the error bars for the figure 9. Variation for resistance can be not that big and it is depended on the error bars as measured for various samples. Adhesion measured in levels is not familiar. Does it mean adhesion has improved and if yes, then why should it improve in compare to pure WEP.

Line 311: “It exhibiting excellent physical and mechanical properties” .

Authors present neither comparison nor explanation why these properties are so excellent.

For the Electrochemical Characterization. It is of importance to present reproducibility data for such analysis of such challenging samples due to possible high aggregation and surfaces containing quite large pores. And again, the trends in the Nyquist diagrams for 15 days and 30 days are questionable, since 0.5% and 1% samples demonstrate quite different behavior and even improved corrosion resistance.

In general, this work has a good potential but requires significant improvement. 

Reviewer 2 Report

I don't have any new queries. Correct the English and the style, and improve the graphics quality

Round 2

Reviewer 1 Report

 Polydopamine Modified Graphene Oxide-TiO2 Nanofiller for Reinforcing Physical Properties and Anticorrosion Performance of Waterborne Epoxy Coatings.

The manuscript was improved a bit in accordance with the reviewers’ comments. However, some aspects still need to be improved. The work has a good potential for a good publication but in my view require improvements.

 About the provided responses:

Response 1. “It is a known fact that due to corrosion caused by petrochemical equipment, oil and gas pipelines, machinery and other infrastructure damage, has caused serious economic losses, waste of resources and environmental pollution”. Corrosion is a fundamental process and a corrosive medium or oxidization in air/gas will lead to the corroded materials. Thus, petrochecmical equipment, oil and gas pipelines can not cause the corrosion phenomenon but are the subject of occurred corrosion due to the corrosive environment. Also, in the sentence it is unclear what has caused economic losses. I guess the occurred corrosion. I would suggest authors to check their manuscript language in some appropriate English correction services.

There are also again several small errors in the new text such as:

Line 303 “a dense Physical barrier”

Line 321 “and improving Interfacial interaction”

Response 2. “the coating thickness was controlled between 100 and 120μm with the help of film applicator.”. Thus, the thickness was not measured but rather is expected to be within range of 100 to 120 um as based on the applicator calibration. It is not fully correct then to say in the text that it was controlled or measured.

Response 3: “It is also found that after function with PDA shows red shifting of the D band peak from 1356cm-1 to 1305 cm−1 and the G band peak shifts from1587cm−1 to 1544cm−1. The red shift of the band mainly comes from two aspects: First, due to the interaction between GO and PDA, it may also be caused by laser heat loss.” The construction of the sentence in English looks strange. What do you mean by laser heat loss? There is a heat transfer to the material and possible surface heat damage.

Response 4: Since nano-filler (nano-PDA@GO-TiO2 composite) is completely coated by resin, it plays a reinforcing role in the spatial structure of resin. Filling the voids of the polymer segment itself, and improving Interfacial interaction of the coating and the substrate. When the coating is impacted, the nanofiller absorbs part of the impact energy to improve the mechanical properties such as adhesion and impact resistance of the coating.

Nano-fillers do not necessary contribute to the improved mechanical properties for nanocomposites. Often it is opposite due to occurred nanoparticle agglomeration. This aspect, as a well dispersion of the nanoparticles or clustering in the matrix was not fully investigated in this research and would require some additional methods with high resolution. About filling the voids, there is a size effect. It is unclear about adhesion, which is a set of intermolecular forces. Usually, adhesion is the force which needs to be applied to separate two surfaces. Thus, do you mean adhesion of the coating and substrate or adhesion of nanofiller with the matrix?

Several reviewer questions were not answered in the response letter. Following questions still need to be addressed:

Line 293: “the micropores gradually decrease and the surface of the PGT-2%/WEP coating (Fig. 8d) is hardly to see the micropores.” I do not think authors have enough support in the presented SEM figures to claim it. Clearly, quite big pores, i.e. several um, are presented on all figures. Also, how representative these few images are? Are these figures were measured in SE or BSE? There is an interesting contrast for the GO/Ti O2 containing coatings, which can suggested that there are big aggregates in the coatings. Thus entire measured properties can significantly vary on the measured areas.

Line 311: “It exhibiting excellent physical and mechanical properties” . Authors present neither comparison nor explanation why these properties are so excellent.

For the Electrochemical Characterization. It is of importance to present reproducibility data for such analysis of such challenging samples due to possible high aggregation and surface containing quite large pores. And again the trends in the Nyquist diagrams for 15 days and 30 days are questionable, since 0.5% and 1% samples demonstrate quite different behavior and even improved corrosion resistance.

Author Response

This manuscript is a resubmission of an earlier submission. The following is a list of the peer review reports and author responses from that submission.

Round 1

Reviewer 1 Report

Polydopamine modified graphene oxide-TiO2  nanocomposites for reinforcing physical properties and anticorrosion performance of waterborne epoxy  coatings.

This manuscript deals with development of waterborne epoxy based coatings for anticorrosion protection with addition of GO-TiO2 fillers. In general, the development and investigation of such coatings are under high research interest and importance.

In the current work authors were trying to prepare such coating and apply many research techniques to investigate its properties. In principle, this work will be very interesting for readers but the quality must be significantly improved. It looks that the authors did succeed on producing required filler nanoparticles and the resulting coatings. However the main and serious issues are:

-          English is not scientifically appropriate and contains way too many errors in words and sentences. For instance: (Introduction: line 36: it must be – coatings have widely been used; line 26: coating industry: Line 42 is hard to understand : restrictions and required and to be strictly increased? ; Line 51: Because of….resin usually contains? It should be rephrased. Also here, a large amount (this word should be added); further in the text to aggregates, it should be to aggregate). And many more examples.

-          A lot of inaccuracies and mistakes. It demonstrates that authors did not pay attention to the manuscript quality. For instance: - Our country – what country to you mean? Is this work interesting only for this country and not for broad readers? Line 48: Waterborne should be waterborne, line 64: .which, should be “. Which” and many more errors like these.

-          Title and introduction are confusing. Nanocomposite is by itself a coating, thus it is confusing that you combine a nanocomposite and a coating together. In fact, in the work you produce GO-PDA-TiO2 nanofillers, i.e. nanoparticles, which you later add to the polymer matrix. No information is provided about produced nanoparticles sizes and their tendency to aggregate.

-          The introduction is rather poor written, for instance there are about 280 papers in literature and many recent ones about waterborne epoxy coatings while authors just discuss a few, there are also many studies on GO-epoxy anticorrosion coatings. Also, the title is very long and don’t really represent the work.

-          There are many techniques which were applied to characterize the studied systems but the exact measurement conditions are not specified. The results don’t have a deep scientific discussion. For instance: XRD, authors just briefly discuss the significant change in the strong GO peak around 12 degree in the PDA-GO-TiO2. Raman: measurement conditions are very important, laser power? Objective and power density? Illumination time? The increase and changes in D and G bands are important, D band is often related to the defects in material. Why no data is measured for pure PDA? No data is measured for the pure epoxy coating and the coating with nanoparticles? XPS: there are clearly peak fittings done, what about the amount of chosen peaks? What do they show? TEM: why no larger scale images are shown for the PDA-GO-TiO2, how representative are these images? Did you check the d spacing for pure GO also?

-          Mechanical characterization of prepared coatings is unclear, not everyone is aware about the mentioned standards. What are the units for adhesion? Any images of prepared coatings?

-          Electrochemical characterization needs a better analysis of data. For instance, the shape of arcs in Nyqvist plots, related to the corrosion mechanisms, is not discussed. The radius is related to the corrosion rate. Why the radius for pure WEP coating increased after 30 days in compare with 15? Why different schemes were used after 15 and 30 days to do Nyqvist analysis. Also, authors should indicate about data reproducibility. As evident from table 1 there is large and strange variation in icorr.

Reviewer 2 Report

The paragraph concerning mechanical properties reports imprecise sentences and has to be re-written.

Many typos are present. Please, revise English style.

Reviewer 3 Report

The work presented by Wang et al, entitled Polydopamine modified graphene oxide-TiO2 nanocomposites for reinforcing physical properties and anticorrosion performance of waterborne epoxy coatings, represents a systematic study on how GO, self-polymerized dopamine and nano TiO2 can improve the protective properties of an epoxy resin in the coating of steel.

Before recommend this manuscript for publication, I have a general concern and some queries. I am wondering what will happen if authors would not employ the polydopamine, and only prepare GO – TiO2 to then mix with the epoxy resin and test the coating. I suggest to prepare this, on my opinion, control sample, characterize it and test its coating performance to enrich and understand better the overall performance of the GO-PDA-TiO2 nanocomposite, with a comprehensive discussion of the new observed features.

Other queries are the following:

·        English must be improve throughout all the manuscript. It is very difficult to follow authors discussion with the actual English redaction style.

·        Introduction: I would avoid political claims, since twice is written the particular efforts of China regarding environmental issues.

Concerning science, it is not clear to me why is necessary the functionalization of GO in order to make it react with an epoxy resin. Authors discussed previous works in where the adding of nanoparticle fillers were enough to improve the properties of the epoxy composite. It is even stranger to me the employment of a reductant. This mean than eliminating GO surface functional groups, the interaction with the resin becomes better? More references and detailed discussion is necessary. I would also suggest to add another scheme (or highlight better the existing one) on how the dopamine is reacting with GO for the shake of clarifying these aspects.

·        Experimental: What is “Tris-HCl” (line 111) and why authors are employing a random Cu K (line 151) radiation to analyse the XRD data? Please explain. In addition, the coating of the steel pieces must be better describe, indicating the precise amount of composite employed.

·        Results and discussion: It is not clear to me the actual reduction of GO by treating with dopamine. On the XRD the 002 peak at 12 o in GO disappeared, but I cannot see the same plane at 26 o on the modified sample diffractogram. Authors only discussed the broad peak at 23.5 o, which has little sense to have an “incomplete” reduction. Once the reduction of graphene oxide is performed, the 260 peak is always merging out (check 10.1016/j.carbon.2012.09.059). In addition, the Raman is not consistent with the reduction, just only with the functionalization. I suggest to clarify the GO’s reduction extent more precisely.

XPS C1s of the functionalize sample is wrongly fitted. The position and FWHM of the bands do not correspond to the pristine GO. Please correct. In addition, C-N groups in C1s have been reported to lay at higher binding energy than 284.8 eV (please, check and cite 10.1016/j.carbon.2018.11.010, 10.1016/j.carbon.2015.09.018)

The sedimentation tests are missing and authors are referring twice to figure 7 (tem and sedimentation). Please, correct and add the proper figure.

I would suggest to explain better the conclusions extracted for the mechanical tests. It is really hard to follow.